# In the Search of an Assessment Method for Urban Landscape Objects (ULOs): Tangible and Intangible Values, Public Participation Geographic Information Systems (PPGIS), and Ranking Approach

**Barbara Sowińska-Świerkosz [1], Malwina Michalik-Śnieżek [2,*], Dawid Soszyński [3]
and Agnieszka Kułak [3]**

[1]   Department of Hydrobiology and Ecosystems Protections, University of Life Sciences in Lublin, Dobrzańskiego 37, 20-262 Lublin, Poland; barbara.sowinska@wp.pl

[2]   Department of Grassland and Landscape Shaping, University of Life Sciences in Lublin, u. Akademicka 13, 20-950 Lublin, Poland

[3]   Institute of Landscape Architecture, The John Paul II Catholic University of Lublin, Konstantynów 1H, 20-708 Lublin, Poland; dawid.soszynski@wp.pl (D.S.); a.k.ulak@wp.pl (A.K.)

*   Correspondence: malwina.sniezek@up.lublin.pl; Tel.: +48-503-508-961

**Abstract:** The effective assessment of urban space must link subjective and objective approaches. The main aim of the paper was to develop and test such a method of assessment in relation to one of the elements of the urban landscape called urban landscape objects (ULOs). The tested method fulfils the following requirements: (1) merges social and expert opinions, (2) analyzes diverse characteristics of urban space, (3) quantitatively presents the results of values assessments, and (4) features the simplicity of structure and ease of public understanding. The method was tested in relation to 34 ULOs located in three different functional sites within Lublin city (Poland). The result enables authors to answer three research questions: (1) How do people perceive ULOs located in different sites? (2) What kinds of tangible values possess different ULOs and how can they be expressed? (3) How can intangible and tangible values be merged? The general finding of the study showed that the Old Town features the highest ranked position in terms of all the values (mean aggregation index (A) ULOs = 0.64), together with the higher share of the most appreciated ULOs, whereas the Lagoon features the lowest ranked position (mean AULOs = 0.35), also statistically comparable with the Campus (mean AULOs = 0.45).

**Keywords:** urban landscape object; urban landscape; public participation geographic information systems (PPGIS); urban areas; urban landscape assessment

## 1. Introduction

Europe is among the world's most urbanized regions, with approximately 73% of Europeans living in urban areas, a figure expected to increase to 80% in 2050 [1]. However, despite such a large population growth in cities, it is expected that they will provide a decent quality of life and contribute to improving people's wellbeing. The valuation of the urban landscape as a main element in the quality of life is crucial from the point of view of spatial planning processes in cities, which should be consistent with the concept of sustainable development, especially in terms of preserving cultural landscapes, as well as directing and harmonizing its changes resulting from social, economic, and environmental processes. This valuation should simultaneously draw upon social impact assessments, by involving the community in decision-making processes [2,3].

In connection with the need to assess urban areas and their role in creating the wellbeing of urban residents and the comfort of urban space users, many authors have attempted to develop and test methods for assessing urban spaces. These methods differ mainly from the point of view of the type of assessed urban space. Some of them refer to the assessment of urban landscapes [4–6], while other methods are focused on assessing the urban environment [7,8]; furthermore, others investigate green urban infrastructure [9–11]. Differentiation can also be seen in the elements under assessment, including visual quality of the urban environment [12,13] and urban ecosystem services [14–16], as well as in relation to intangible ecosystem services such as cultural ecosystem services (CES) [17,18]. Regarding the similarities, most authors agree that consideration of the opinion of different stakeholders and the illusion of different driving factors and structural, functional, and social indicators are crucial for the elaboration of effective assessment methods.

To date, many publications have been provided in which the authors take up the problem of assessing urban spaces. These assessments are based on two types of approaches: (1) objective (expert), using specialists' knowledge and experiences, as well as mathematical techniques and data, recently related mainly to the use of geographic information systems (GIS), or (2) subjective, based on the individual feelings of city dwellers. Among the various objective methods mentioned is the visual protection surface (VPS), allowing the exploration of geometrical relationships between the scope of visual protection of the city and the maximum heights of new buildings [19]. GIS analysis, on the other hand, is based on the use of landscape metrics as a tool of describing, quantifying, and monitoring the spatial configuration of urbanization at landscape levels [20–22], as well as the multifactor analysis of cities' landscape quality based on the use of spatial, ecological, cultural, and visual indicators [23]. These methods are based on the use of a wide range of indicators, both qualitative and quantitative, providing the possibility of an objective presentation of various characteristics of urban space. Even these methods, however, are not wholly objective as they require the participation of an expert who will properly interpret the results.

Methods based on the subjective approach most often relate to the issue of the quality of life and visual assessment of the urban landscape. One of the first and most important methods of this type was published by Kevin Lynch in his work "The image of the city" [24]. The author analyzed the visual quality of American cities by studying their citizens' mental image, while, in another publication, he synthesized urban quality according to five main aspects defined from citizens' point of view: vitality, sense, fit, access, and control [25]. It should be emphasized that city image can be different in different stakeholders' visions, such as the local community and visitors. This was proven by papers in relation to different countries [26–28]. Currently, the most common methods for carrying out this type of assessment are questionnaires and survey campaigns. The structure and form of questionnaires vary greatly in their different approaches to evaluation. Some use willingness-to-pay and willingness-to-accept questions [29], while others use preference questions or grades, i.e., 1–5. The increasingly common method of conducting surveys to obtain information about subjective feelings or preferences regarding space development is mapping spatial attributes [30], as well as geo-questionnaires [31]. These tools are also called public participation geographic information systems (PPGIS). If they are properly designed for lay users and selected for the problem at hand, they encourage the participation of inhabitants in the process of urban planning and help eliminate possible conflicts between particular groups of interest [32,33]. The geo-questionnaire provides information characterizing the respondents and their preferences. With the use of an interactive map, the respondent can use three types of objects, namely, point, line, and polygon, to mark the sites relevant to a given question. Moreover, the respondents can assess and determine the quality of a given place or object [33].

Describing urban landscapes in a comprehensive manner, however, requires the use of a wide range of approaches, methods, and indicators. The effective assessment of urban space must link the subjective and objective approaches; management practices must simultaneously consider the opinion and needs expressed by the general public and be supported by expert knowledge under

existing institutional conditions. Using this approach, an integrated view of the urban space may be obtained as a complex ecological, cultural, social, and visual system [34]. Moreover, to have policy and management capability, a combination of social and expert opinions should feature the high applicability reflected by the simplicity of structure and ease of public understanding [35].

In this paper, the authors attempt to adopt such a method linking social and expert opinions in relation to one of the elements of urban space, called urban landscape objects (ULOs). These objects are partially understood as analogous to landmarks, defined as an external point of reference that helps orientation within a familiar or unfamiliar environment [36]. However, unlike landmarks, ULOs are treated only as objects physically located in the landscape, ignoring the features of these objects or symbols located on objects that can be treated as landmarks. In this study, ULOs are understood to be natural, seminatural, or anthropogenic objects that can be identified easily in the landscape. They represent visual, functional, and finite units which can be specified, such as a tree, rock, or fountain which can be named, such as the Blue Beach, the Main Square, or the National Museum. The assessment of ULOs is an important element in the analysis of urban spaces and is reflected in the planning of these spaces in accordance with the needs of residents and the requirements of sustainable development. The research conducted so far to assess urban landmarks most often concerned their visual aspects [37]. Taking into account their impact on the urban landscape quality, however, next to their aesthetic features, other values related to the intangible dimensions should be analyzed [34]. Such perceptual understanding of inter alia city landscape was emphasized in the "European Landscape Convention" [38] which has changed the concept of the landscape from the conventional meaning used by geographers and landscape ecologists to include the subjective dimension deriving from peoples' perception [39]. Neglect of the this dimension can have serious implications, since, despite the adoption of an environmentally friendly and income-generating model of space management, aspects that are crucial for social coherence, such as a sense of belonging and local tradition, will not be taken into account [40]. Especially today, when the global trend in urban planning is moving toward smart city development, the illusion of intangible dimension is needed to create more effective integration of the changes within each urban community [41]. Moreover, material and nonmaterial values of city are not isolated; rather, they are intertwined and form a complex social–ecological system [29,42]. The assessment of these values should reflect not only the subjective feelings and preferences of the users of these objects, but also objective assessments that will not be affected by random assessment factors [34,43]. The merging of subjective and objective points of view would provide alternative ways of seeing the same object in different contexts [44], thus presenting an image of a city in a comprehensive manner [45].

As such, the main aim of the paper was to develop and test a method of ULO assessment which fulfils the following requirements: (1) merges social and expert opinions; (2) analyzes diverse characteristics of urban space, including aesthetic, symbolic, educational, spiritual, entertainment, enjoyment, heritage, and universal values; (3) quantitatively presents the result of value assessments enabling comparisons among different types of sites, objects, and values; (4) features the simplicity of structure and ease of public understanding.

The method, based on the integration of the objective and subjective approaches in landscape assessment, was tested in relation to Polish conditions because of a few reasons. Firstly, the state of social dialogue and the level of public participation in all parts of Poland are low. Polish law allows for participation in urban planning procedures, but not everyone wants to exercise this right. This is due to the weakness of the nongovernmental organization (NGO) sector, inadequate local planning information sources, and low activity of local communities [46,47]. In all polish cities and communes, meetings and workshops where the opinions of residents are collected are usually organized in a specific place and time; thus, not everyone can participate in them. Our method allows collecting opinions and evaluations via the Internet. Therefore, all social groups can express their opinion. The presented method was used in relation to the city of Lublin (eastern Poland, Lubelskie voivodeship),

but it can also be applied to other cities in Poland and other parts of the world with similar problems of low social involvement.

## 2. Materials and Methods

### 2.1. Study Site

The method was applied to three test areas located in Lublin City, eastern Poland (Lubelskie voivodeship), which significantly differed in the characteristics of their landscape (Figure 1). These areas were selected on the basis of interviews conducted with random students attending Lublin universities. In order to identify study areas, students were selected due to the academic nature of the city, in which as many as 28% of residents are students [48]. In response to a request mentioning several areas in the city that are characterized by different landscapes, students most often indicated the Old Town (65.2 ha), the Campus (171.6 ha), and the area surrounding the city's Lagoon (1583.3 ha). The first site is characterized by the presence of historical buildings and sites, which mainly serve as restaurants, bars, and art galleries; it is typified by a high level of anthropogenic transformation and a high degree of aesthetic harmony. The second site, the Campus, is composed of buildings and sites serving various functions, such as student halls of residence, canteens, student clubs, sports fields, and a park. The city's Lagoon is surrounded by three kinds of areas: forests and peat bogs, a recreational complex with a beach, pools, and a rope park, and an estate of detached houses. Such diversity of landscape types and terrain functions allowed us to test the reliability of the method in relation to different kinds of areas.

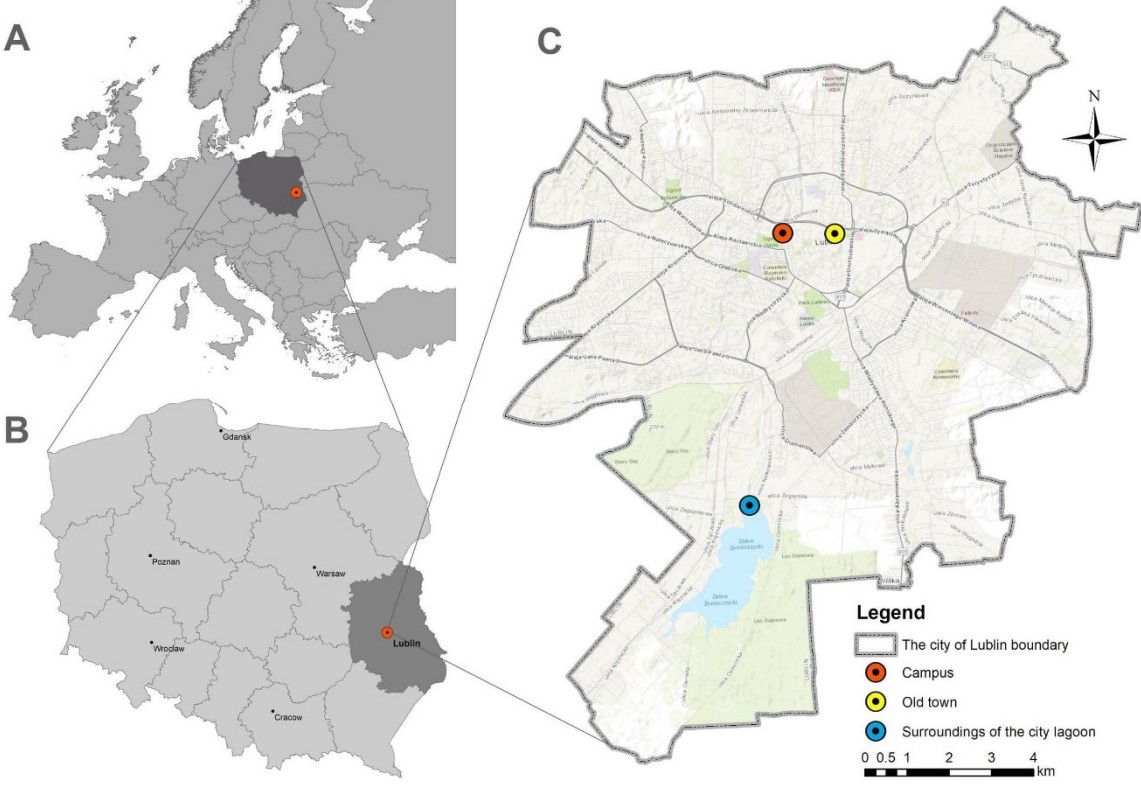

**Figure 1.** Location of study areas within: (**A**) Europe; (**B**) Poland; (**C**) Lublin city border.

### 2.2. Research Procedure

The first stage of the research aimed to select ULOs recognized by the students in relation to the three test areas. A topographic map and orthophotomap of each site were displayed in ArcGIS online with the borders of the three test areas. As the respondent group was chosen, students from two Lublin

Universities (University of Life Sciences in Lublin and The John Paul II Catholic University of Lublin) attended different courses of life sciences. They were asked to mark and name urban landscape objects, located in the areas defined, that they visited at least once. Students were to use the label tool and could mark and name (or only mark if they did not know the name of an ULO) as many objects and sites as they wished. Respondents were previously informed what ULOs were, according to the definition provided in the introduction. Interviews with between 10 and 20 students per session (six sessions in total) were conducted in rooms at the university during June 2019, lasting approximately 15 min. A group of 80 students in total took part in the research. Second, the total list of students' responses was examined; those objects marked by at least 10% of the respondents were included in further stages of the analysis (i.e., 34 objects). The second stage of the research aimed to analyze different types of values possessed by ULOs on the basis of two research questions referring to the subjective and objective perspectives (Figure 2).

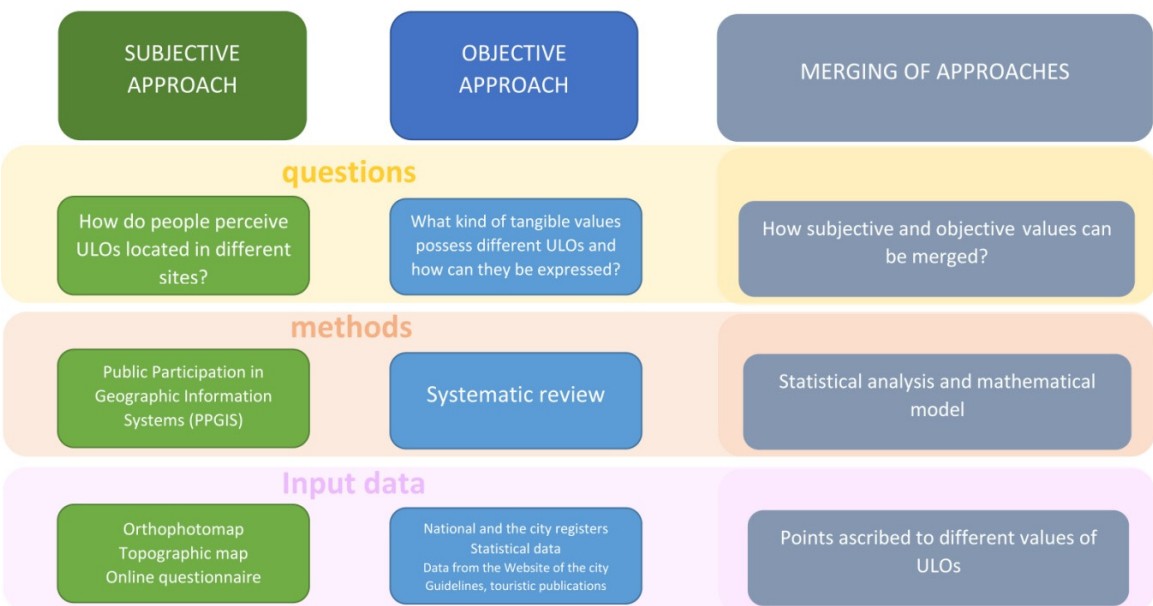

**Figure 2.** Methodological framework based on three research questions.

The basis of the assessment framework used to answer these two questions was derived from the results of a systematic review on aspects related to urban objects' values. The review comprised two stages. Firstly, the English literature review was conducted within the Scopus database (2019) using keywords such as (cultural) ecosystem services, urban (city) assessment, urban (city) objects, and urban (city) values to detect which values are of a tangible character and can be expressed objectively. Here, the Common International Classification of Ecosystem Services (CICES) framework V5.1. for the assessment of ecosystem services was the most adequate. This classification was selected as it has been widely used as the basis of mapping landscape values across different regions and countries [49] and has strong legal and methodological basis, as it was developed out of the work on environmental accounting undertaken by the European Environment Agency (EEA). This classification divides cultural values (services) into two general groups: (1) direct, in situ, and outdoor interactions with living systems that depend on presence in the environmental setting considering as intangible values in our study, and (2) indirect, remote, often indoor interactions with living systems that do not require presence in the environmental setting considering as tangible values in our study. As ULOs can be both abiotic (buildings) and biotic (greenery, water), the following interactions and characteristic were considered in the research:

- Characteristics (biotic) that enable active physical interactions, characteristics that enable activities promoting health, recuperation, or enjoyment, and elements (biotic) used for entertainment—assessed as enjoyment intangible values and entertainment intangible values
- Characteristics (abiotic) that enable intellectual interactions and characteristics that enable education and training—assessed as educational intangible values
- Characteristics or features (abiotic) that have an existence, option, or bequest value and characteristics (biotic) that are resonant in terms of culture or heritage—assessed as universal intangible values and heritage intangible values
- Spiritual interactions (abiotic) and elements (biotic) that have sacred or religious meaning—assessed as spiritual intangible values
- Characteristics (abiotic) that enable symbolic interactions and elements (biotic) that have symbolic meaning—assessed symbolic intangible values
- Characteristics (abiotic) that enable active or passive experiential interactions and characteristics (biotic) that enable aesthetic experiences—assessed as aesthetic intangible values

The detailed criteria of value assessment (questionnaire structure and objective grading scale) were partially based on the results of the review of literature [50–52]. The final structure of the framework was derived from the local context: the availability/ lack of data for objective criteria and the characteristics of ULOs detected by respondents.

The first question dealt with the issue of the public perception of ULOs: "How do people perceive ULOs located in different sites?" Consequently, the PPGIS method was used to answer this question. As defined in the previous stage, 34 elements were included in the online questionnaire, based on the Survey 123 ArcGIS application, aimed at discovering students' opinions on the intangible values of ULOs. The questionnaire was completed by the group of 50 students who took part in the first stage of the research and who had visited the assessed ULOs at least once. Respondents' profiles are illustrated in Table 1. The questionnaire contained personal metrics and four questions related to each of the 34 assessed objects. Each landscape object was presented by name, location, and photograph (Figure 3B). Students expressed their preferences using a four-point scale (Table 2A) and were asked how they perceived a given landscape object as being visually attractive (aesthetic value), suitable for contemplation, meditation, or prayer (spiritual value), a source of enjoyment (enjoyment value), and worthy of preservation (universal values). We used the mean value of points given by all of the respondents to produce a final value. The physical conditions of the research setting were similar to those of the first stage.

**Table 1.** Profile of the respondents (students who express their preferences on intangible values of urban landscape objects (ULOs)).

| Age | 20–23 years old | 100% |
|---|---|---|
| Gender | Female | 73% |
| | Male | 27% |
| Origin | Village | 55% |
| | Small-size city (up to 10,000 inhabitants) | 8% |
| | Medium-size city (10,000–50,000 inhabitants) | 16% |
| | Large-size city (more than 50,000 inhabitants) | 21% |
| Period of stay in Lublin | I do not stay in/commute to Lublin everyday | 20% |
| | 1–5 years | 53% |
| | More than 5 years | 27% |

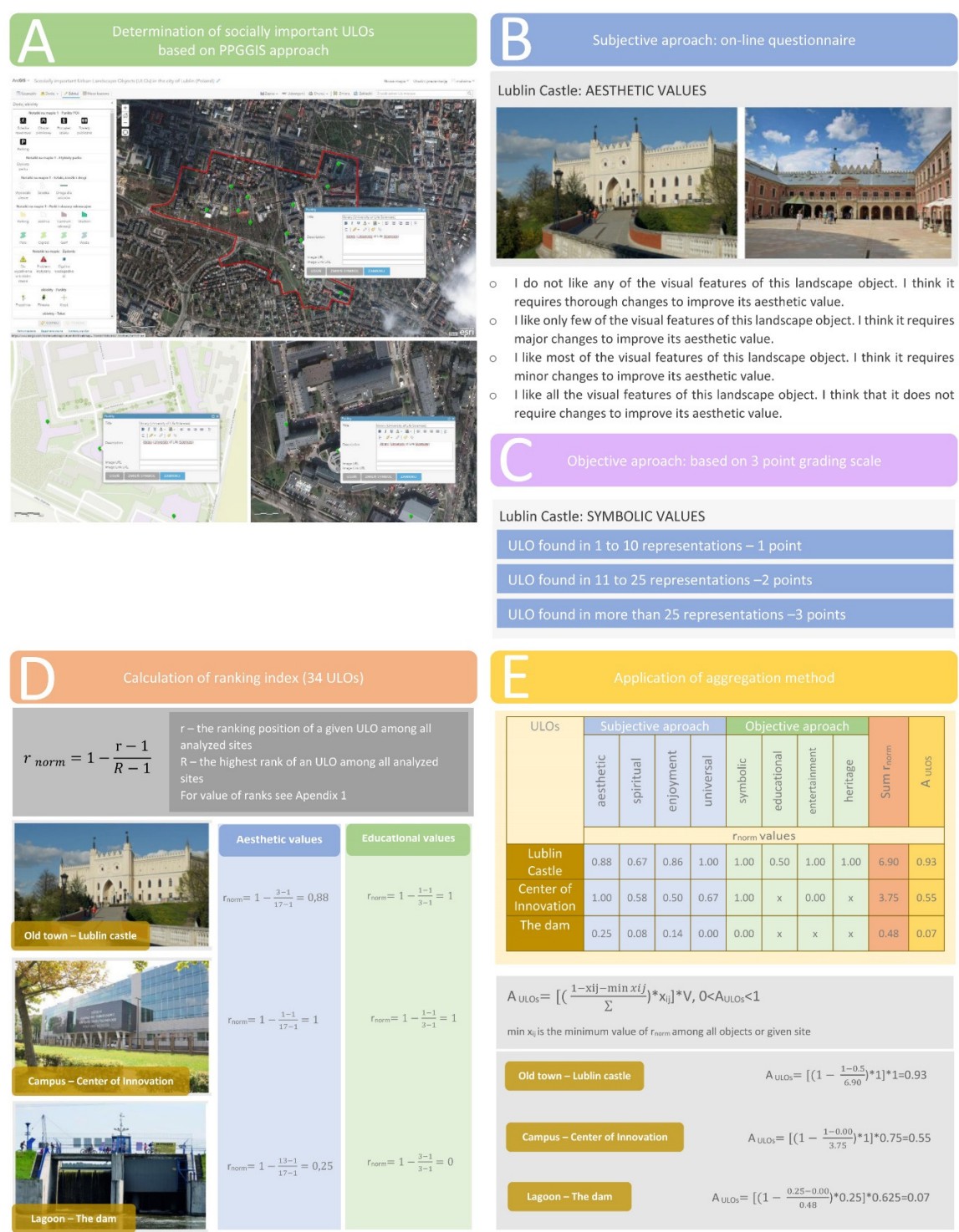

**Figure 3.** Methodological framework illustrated by the examples of three analyzed ULOs located in different sites: Lublin castle (Old town), Center of Innovation (Campus), and the dam (Lagoon).

To answer the second research question ("What kinds of tangible values are possessed by different ULOs and how can they be expressed?"), the objective grading scale was applied in relation to four value types: symbolic, educational, entertainment, and heritage. Symbolic values were based on the number of given ULO representations in commercial visual arts based on seven guides of Lublin from 1901–2014. Educational values were derived from the potential educational function of a given ULO. Entertainment values were expressed objectively by the number of cultural events taking place in a

given ULO per year according to the official website of Lublin City and official websites of particular landscape elements. A protected status was selected to express heritage values. Experts (authors) ascribed grading points to each landscape element using a three-point scale (Table 2B; Figure 3C).

**Table 2.** Ranking assessment criteria: a four-point scale adopted in the assessment of tangible and intangible values of ULOs. PPGIS, public participation geographic information systems; UNESCO, United Nations Educational, Scientific and Cultural Organization.

| A. Questionnaire Implemented to Assess the Intangible Values of ULOs: Subjective Approach Based on PPGIS Method | B. Ranking Scale Adopted to Analysis Tangible Values of ULOs: Objective Approach Based on Systematic Review |
| --- | --- |
| **Aesthetic Values** | **Symbolic Values** |
| Respondents' opinion on the aesthetic values of a landscape object. 0—None of this landscape's objects are visually attractive to me. I think that these elements need fundamental changes aimed at improving their aesthetic values. 1—Only a few of the features of this landscape's object are visually attractive to me. I think that these elements need major changes aimed at improving their aesthetic values. 2—The overwhelming majority of this landscape's object features are visually attractive to me. I think that these elements need only cosmetic changes aimed at improving their aesthetic values. 3—All of this landscape's objects are visually attractive to me. I think that these elements need no changes aimed at improving their aesthetic values. | Grading scale adopted by experts: 0—an object was not included in representations 1—an object found in 1 to 10 representations 2—an object found in 11 to 25 representations 3—an object found in more than 25 representations according to the 7 guides of Lublin from years 1901–2014 |
| **Spiritual Values** | **Educational Values** |
| Respondents' opinion on the sacred and religious services provided by a landscape object. 0—In my opinion, this place is unsuitable for contemplation, meditation, or prayer. I have never done it here, and I am not going to do this. 1—In my opinion, this might be an adequate place to contemplate, meditate, or pray. I have never done this here, but I could try to do it. 2—In my opinion, this place is adequate to contemplate, meditate, or pray. I sometimes did/do this in this place or, if I would like to contemplate, meditate, or pray, I would willingly choose this place. 3—In my opinion, this is a very good place to contemplate, meditate or pray. I did/do this in this place many times or, if I would like to contemplate, meditate, or pray, I will first choose this place. | Grading scale adopted by experts: 0—landscape object of no educational function 1—landscape object of the potential educational function (no educational infrastructure, but you can learn something just by observing) 2—landscape object of secondary educational function (having information boards, construction plans, historical descriptions, etc.) 3—landscape object of major education function (e.g., museum, regional room) |
| **Enjoyment Values** | **Entertainment Values** |
| Respondents opinion on the existing services provided by a landscape object. 0—I do not enjoy this landscape object. 1—I rather enjoy this landscape object. I could go there from time to time. 2—I enjoy this landscape object. I would like to go there often. 3—I enjoy this landscape object. I would like to go there as often as possible. | Grading scale adopted by experts: 0—landscape object that is not a place of any cultural events 1—landscape object being the place of 1 or 2 cultural events per year * 2—landscape object being the place of 3 to 5 cultural events per year * 3—landscape object being the place of more than 5 cultural events per year * * according to the official website of the Lublin City and official websites of particular landscape object according to Getz typology of planned events [53] |

**Table 2.** *Cont.*

| A. Questionnaire Implemented to Assess the Intangible Values of ULOs: Subjective Approach Based on PPGIS Method | B. Ranking Scale Adopted to Analysis Tangible Values of ULOs: Objective Approach Based on Systematic Review |
|---|---|
| **Universal Values** | **Heritage Values** |
| Respondents opinion on the willingness to make efforts to preserve a landscape object.<br>0—I think that this landscape object does not necessarily have to be preserved.<br>1—I think this landscape object should be preserved, but I am not ready to pay for it or devote my spare time.<br>2—I could pay less than 20 PLN (5 EUR) and/or devote less than 1 h to preserve this landscape object.<br>3—I could pay more than 20 PLN (5 EUR) and/or devote more than 1 h to preserve this landscape object. | Grading scale adopted by experts:<br>0—landscape object that is not legally protected<br>1—landscape object in municipal register *<br>2—landscape object in the registry of monuments **<br>3—landscape object being the part of a UNESCO World Heritage site ** |

\* according to the registry of Lublin community. \*\* according to the registry of The National Heritage Board of Poland.

The third question relates to the problem of merging points on the basis of the subjective and objective approaches. As a solution, the two-stage mathematical formula was proposed. Firstly, the value ranking index (Equation (1)) of a given ULO was calculated, taking into account all analyzed ULOs and sites (Figure 3D).

$$r_{norm} = 1 - \frac{r-1}{R-1}, \tag{1}$$

where $r$ is the ranking position of a given object among all analyzed sites, $R$ is the highest rank of element among all analyzed sites, $r_{norm} = 1$ indicates that a given object possesses the highest ranking position among all analyzed sites, and $r_{norm} = 0$ indicates that a given object possesses the lowest ranking position among all analyzed sites.

Next, we applied the aggregation method (Figure 3E) in reference to the ranking positions obtained for all ULOs of a given site using Equation (2), taking into account four variables: (1) the difference between the minimum and maximum ranking position, indicating whether it is a ULO of a specialized function (e.g., tourism, nature conservation, food provision), (2) the sum of the ranking positions of ULOs belonging to a given site, indicating the relative value of a site in terms of its values, (3) the value of the maximum ranking position, indicating whether a given site contains a ULO that possesses outstanding values of a given type, and (4) the V index, which takes into account the share of tangible value types which a given ULO does not possess (e.g., if a given ULO is not under legal protection, its heritage values equal 0).

$$A_{ULO} = \left[ \left( 1 - \frac{x_{ij} - min\ x_{ij}}{\sum_{i=1}^{r} x_{ij}} \right) * x_{ij} \right] * V, \ 0 < A_{ULO} < 1, \tag{2}$$

where *max $x_{ij}$* is the maximum value of $r_{norm}$ among all ULOs of a given site, *min $x_{ij}$* is the minimum value of $r_{norm}$ among all ULOs of a given site, $V = 0.125$ (*1 divided by total number of value types, i.e., 8*), multiplied by the number of value types which a given ULO possesses (e.g., $V = 0.875$ ($0.0125 \times 7$) if a given landscape element is a provider of any seven of the eight analyzed value types), $A_{ULO} = 1$ indicates that a given site possesses the highest ranking position in terms of its values among all sites analyzed, and $A_{ULO} = 0$ indicates that a given site possesses the lowest ranking position in terms of its values among all sites analyzed.

*2.3. Statistical Analysis*

The results were analyzed using Statistica software. In the case of normally distributed data, one-way analysis of variance (ANOVA) was used to define the differences among different types of values ascribed to ULOs and among the three test areas. Statistical differences between means

were measured using the Tukey post hoc test. When the test of normality showed that the data were not normally distributed ($p < 0.00001$), the Kruskal–Wallis test was applied to indicate statistical significance between value types ($\alpha = 0.05$) and the Dunn post hoc test, further adjusted by the Holm FWER (Familywise Error Rate) method, was applied to indicate the values that significantly differed from the rest of the group.

## 3. Results

### 3.1. Results of PPGIS Application

In the first part of the research, using the ArcGIS online application, respondents distinguished 147 different landscape elements, including 61 located in the Old Town, 72 on the Campus, and 14 near the Lagoon. The number of objects identified ranged between five and 55 (mean: 21). Among these objects, 34 were identified by at least 10% of the respondents (Appendix A; Figure 4). Landscape objects selected in the city (16 objects) were diverse in terms of function, including monuments, churches, pubs, and restaurants, as well as objects accompanying them (e.g., stairs). The majority of the 14 locations on the Campus were university buildings. Sports facilities and the academic park were also included in the later stages of this analysis. The Lagoon was only represented by four landscape objects, all of which, except for a dam, were tourist infrastructure elements.

### 3.2. Results of Assessments from Students and Experts: Differences in Mean Values of Points Attributed to Each Type of Value Regardless of the Site Type

The points ascribed by students and experts to all analyzed ULOs (Appendix A) were found to be significantly higher in the case of aesthetic, entertainment, and educational values and lower in the case of heritage values (Table 3A). While analyzing only the intangible values, it was identified that the students attributed a relatively high rank to the aesthetic values, whereas the rank given to the spiritual, enjoyment, and universal categories were of lower significance (Table 3B). Among the tangible values, clear differences were seen between entertainment and other types of values (Table 3C). There were no differences, however, between points given to all four intangible and all four tangible values ($p < 0.05$; $n = 36$; $t$-value = 0.501; $p$-value = 0.308).

### 3.3. Results of Assessments from Students and Experts: Differences in Mean Values of Points Attributed to Each Type of Values in Relation to Each Site Type

The statistical differences among points ascribed to aesthetic, spiritual, universal, and entertainment values of ULOs located in different sites showed that the results were not significant at $p < 0.05$ (Table 4; Figure 5). Heritage values occurred as typical only for the landscape objects located in the Old Town ($p < 0.0001$). Additionally, the latter areas possessed significantly higher enjoyment ($p = 0.022$) and symbolic ($p = 0.005$) values. On the other hand, the Campus featured significantly higher educational values ($p = 0.023$).

### 3.4. Results of Assessments from Students and Experts: Differences in Mean Values of Points Attributed to Values of ULOs Located in Each of the Three Sites Analyzed

When analyzing each site separately, the Old Town ($p < 0.05$; $n = 64$; $t$-value = 1.185; $p$-value = 0.119) and Campus ($p < 0.05$; $n = 56$; $t$-value = 0.746; $p$-value = 0.229) featured no statistically significant differences between the points ascribed to intangible and tangible values. The points ascribed to intangible values at the Lagoon, on the other hand, were statistically higher than tangible ones ($p < 0.05$; $n = 16$; $t$-value = 2.938; $p$-value = 0.003). When analyzing each value type within sites, the differences among the mean values attributed to ULOs were statistically significant in the case of all three sites (Tables 5 and 6). The Old Town featured higher aesthetic and entertainment values than spiritual ones ($p = 0.040$ and $p = 0.048$, respectively), and the Lagoon possessed higher aesthetic than heritage values ($p = 0.011$) (Table 6). The Campus stood out, with its educational and aesthetic values being higher

than most other value types (Table 7), whereas its heritage values were lower than its other values (except symbolic).

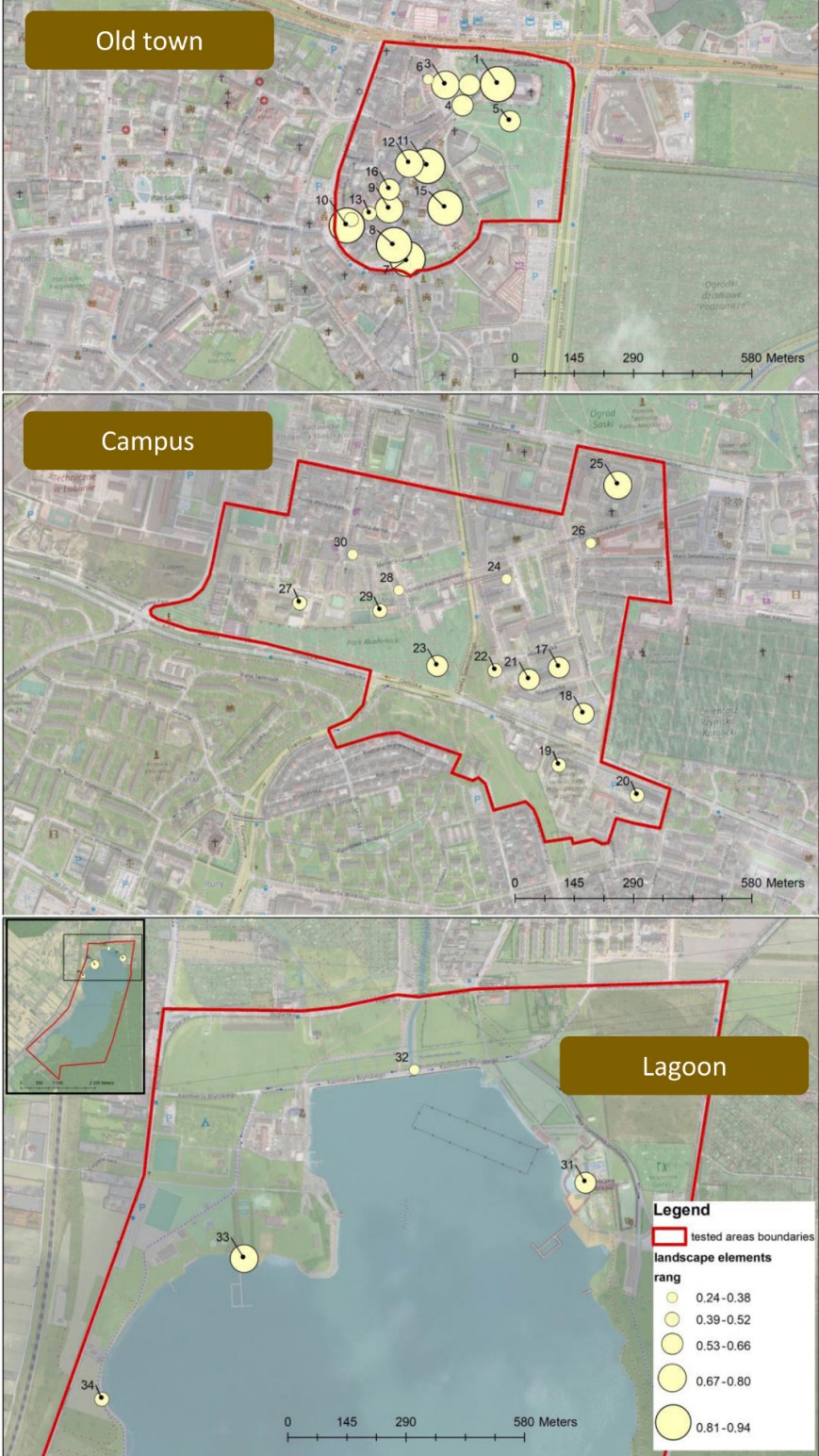

**Figure 4.** Results of ULO mapping, taking into account ranking positions and hotspots. Ordinal numbers of rated objects are marked on the maps. The size of the symbol results from the ranks of the objects that were calculated according to Equation (2).

**Table 3.** Results of Kruskal–Wallis (K–W) test together with Dunn post hoc test further adjusted by the Holm (H) FWER method (levels: value type to ULOs in all sites).

| Values' Types | *p*-Value * | Values' Types | *p*-Value * | Values' Types | *p*-Value * |
|---|---|---|---|---|---|
| **A.** All values' types K–W test: *n* = 272; d.f. = 7; H = 51.78; *p* < 0.00001 | | **B.** Intangible values K–W test: *n* = 136; d.f. = 3; H = 39.91; *p* < 0.00001 | | **C.** Tangible values K–W test: *n* = 136; d.f. = 3; H = 24.58; *p* = 0.00002 | |
| Aesthetic vs. spiritual/enjoyment/heritage/symbolic | <0.0001 | Aesthetic vs. spiritual/enjoyment/universal | <0.0001 | Entertainment vs. educational | 0.521 |
| Entertainment vs. spiritual | 0.015 | Universal vs. spiritual | 0.041 | Entertainment vs. heritage | <0.0001 |
| Entertainment vs. enjoyment | 0.155 | Universal vs. enjoyment | 0.348 | Entertainment vs. symbolic | 0.011 |
| Entertainment vs. heritage | <0.0001 | - | | Entertainment vs. heritage | 0.001 |
| Entertainment vs. symbolic | 0.014 | - | | Educational vs. symbolic | 0.055 |
| Educational vs. heritage | 0.001 | - | | Symbolic vs. heritage | 0.362 |
| Educational vs. enjoyment | 0.669 | - | | - | |
| Educational vs. spiritual | 0.099 | - | | - | |
| Educational vs. heritage | 0.06 | - | | - | |
| Educational vs. symbolic | 0.0944 | - | | - | |
| Universe vs. heritage | 0.602 | - | | - | |

* *p*-Value mean post hoc test value.

**Table 4.** Results of the Kruskal–Wallis test ($p < 0.05$, $n = 34$; d.f. = 2) together with Dunn post hoc test further adjusted by the Holm FWER method (levels: value types to ULOs among different sites); n.s., not significant; s., significant.

| | Kruskal–Wallis Chi-Squared Statistic | *p*-Value | Significance | Sites | *p*-Value |
|---|---|---|---|---|---|
| Aesthetic | 3.943 | 0.139 | n.s. | - | - |
| Spiritual | 0.905 | 0.636 | n.s. | - | - |
| Enjoyment | 7.645 | 0.022 | s. | Old town vs. Campus<br>Old town vs. Lagoon | 0.019<br>0.393 |
| Universal | 3.138 | 0.208 | n.s. | - | - |
| Educational | 7.050 | 0.023 | s. | Campus vs. Old town<br>Campus vs. Lagoon<br>Old town vs. Lagoon | 0.213<br>0.022<br>0.123 |
| Heritage | 22.055 | <0.0001 | s. | Old town vs. Campus/Lagoon | <0.0001 |
| Entertainment | 1.272 | 0.529 | n.s. | - | - |
| Symbolic | 10.418 | 0.005 | s. | Old town vs. Campus<br>Old town vs. Lagoon<br>Campus vs. Lagoon | 0.006<br>0.142<br>0.826 |

*3.5. Results of Aggregation Procedures: Differences in AULO (Agregation Index of Urban Landscape Objects) Values in Relation to Each of the Three Sites Analyzed*

By applying ranking and aggregation procedures, we revealed that significant differences between AULOs existed among each of the three sites analyzed ($p = 0.036$) (Table 8). The Old Town featured the highest total values (mean AULO = 0.64) comparable with other sites ($p = 0.063$), and the Lagoon featured the lowest total values (mean AULO = 0.35), also comparable with the campus ($p = 0.434$). Moreover, we found that, among all of the ULOs analyzed, the most highly rated were those located in the Old Town: Lublin Castle, the Dominican Church, and Archcathedral (AULOs = 0.93, 0.92, and 0.92, respectively), followed by the Fara Square (AULO = 0.92) and Trinity Tower (AULO = 0.89) (Appendix A). This area, however, also featured the element with the second lowest AULO value, an ice cream shop near the castle (0.10). While analyzing the Campus, the highest AULO value was obtained for the main building of the Catholic University of Lublin (0.77) and, surprisingly, the lowest was obtained the Main building of the Marie Curie-Skłodowska University (0.20). Of the Lagoon landscape elements, the Marina together with the bicycle path was ranked the highest (0.45) and the dam was ranked the lowest (0.07). As a result, The Old Town included objects of diverse values (SD = 0.259), whereas those landscape objects located within the Lagoon (SD = 0.161) and Campus (SD = 0.182) had similar levels of values.

**Table 5.** ANOVA for factor groups (levels: value types in each site—Old Town, Lagoon).

| | The Old Town | | Lagoon | |
|---|---|---|---|---|
| | $n = 128$; $\alpha = 0.05$; $F_{crit} = 2.103$; $F_{7,120} = 2.42$; $p = 0.023$ | | $n = 32$; $\alpha = 0.05$; $F_{crit} = 2.514$; $F_{7,24} = 3.124$; $p = 0.017$ | |
| | Mean | Median | Mean | Median |
| Aesthetic | 2.02 | 2.10 | 1.65 | 1.65 |
| Spiritual | 1.15 | 1.00 | 1.18 | 1.30 |
| Enjoyment | 1.38 | 1.40 | 1.10 | 1.10 |
| Universal | 1.47 | 1.45 | 1.23 | 1.20 |
| Educational | 1.56 | 1.50 | 0.75 | 1.00 |
| Heritage | 1.44 | 2.00 | 0.00 | 0.00 |
| Entertainment | 2.00 | 2.50 | 1.25 | 1.00 |
| Symbolic | 1.69 | 2.00 | 0.50 | 0.50 |

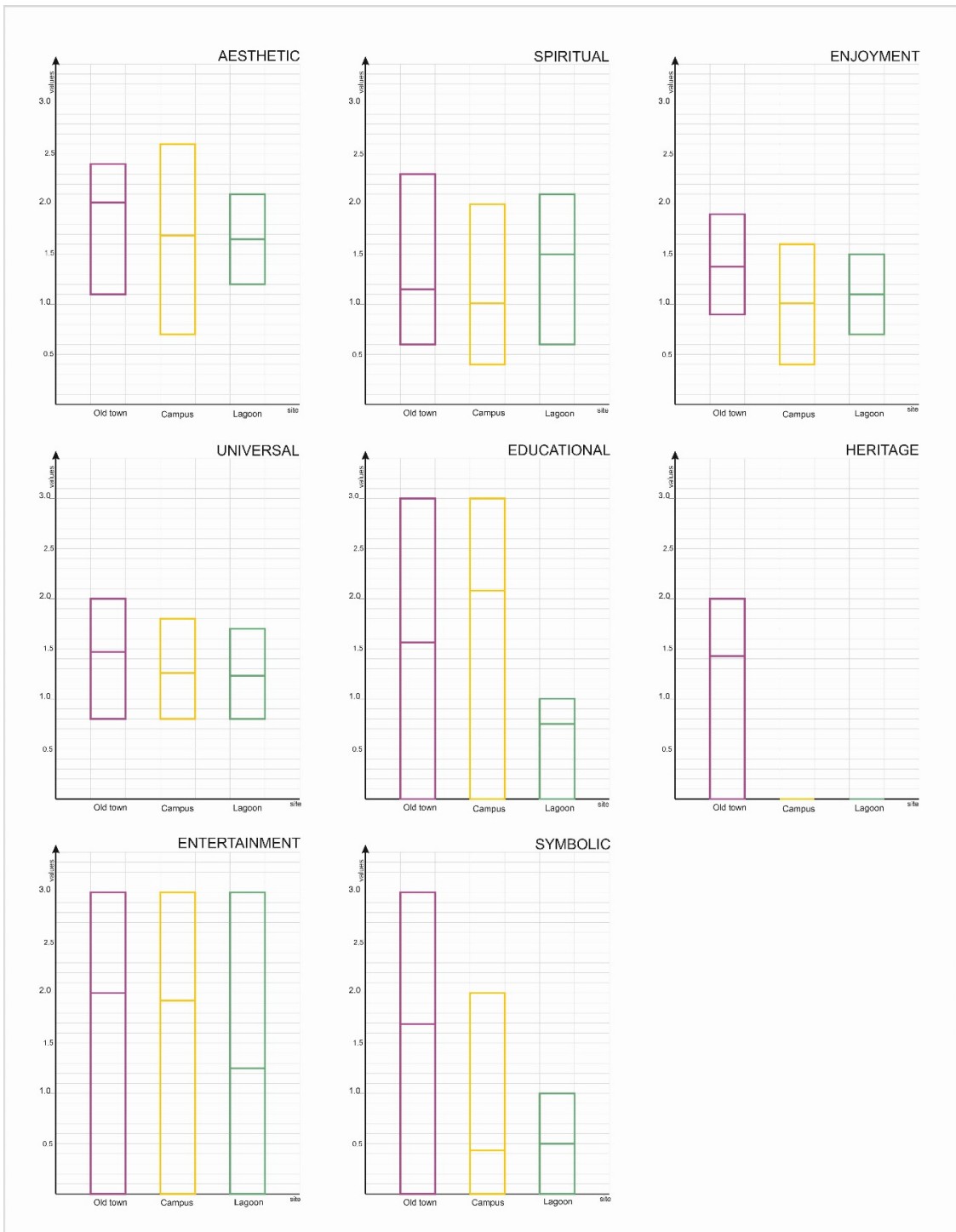

**Figure 5.** Maximum, minimum, and mean values given to each of the three sites analyzed.

**Table 6.** Results of the Tukey post hoc test (levels: value types in each site—Old Town, Lagoon).

| The Old Town | | | Lagoon | | |
|---|---|---|---|---|---|
| Levels | Q Statistic | *p*-Value | Levels | Q Statistic | *p*-Value |
| Aesthetic vs. spiritual | 4.4738 | 0.040 | Aesthetic vs. heritage | 5.637 | 0.011 |
| Entertainment vs. spiritual | 4.3772 | 0.048 | - | - | - |

**Table 7.** Results of the Kruskal–Wallis test together with Dunn post hoc test further adjusted by the Holm FWER method (levels: value types in each site—Campus).

| Values' Types | p-Value |
|---|---|
| $n = 112$; d.f. $= 7$; H $= 55.198$; $p < 0.0001$ | |
| Aesthetic vs. spiritual | 0.474 |
| Aesthetic vs. entertainment | 0.675 |
| Aesthetic vs. heritage | <0.001 |
| Aesthetic vs. symbolic | 0.003 |
| Educational vs. spiritual | 0.196 |
| Educational vs. enjoyment | 0.323 |
| Educational vs. entertainment/symbolic/heritage | <0.001 |
| Spiritual vs. heritage | 0.045 |
| Spiritual vs. entertainment | 0.493 |
| Universal vs. heritage | 0.0006 |
| Universal vs. entertainment | 0.691 |
| Universal vs. symbolic | 0.805 |
| Entertainment vs. symbolic | 0.004 |
| Entertainment vs. heritage | 0.023 |

**Table 8.** Results of the Kruskal–Wallis together with Dunn post hoc test further adjusted by the Holm FWER method (levels: Agregation index of ULOs ($A_{ULOs}$) among sites).

| Values' Types | p-Value |
|---|---|
| $n = 34$; d.f. $= 2$; H $= 6.617$; $p = 0.036$ | |
| Old town vs. Campus/Lagoon | 0.063 |
| Campus vs. Lagoon | 0.434 |

## 4. Discussion

### 4.1. Differences among the Three Sites Analyzed

The general finding of the study showed that the Old Town featured the highest ranked position in terms of values, together with the higher share of the most appreciated ULOs (Table 9). This finding is not surprising as this area is multifunctional and has the longest history of human use, resulting in the existence of historic buildings of high visual values. Other studies on managing historic cities also showed that city centers feature unique social and universal values [54,55]. For example, in a study conducted by Atanur et al. [56], it was discovered that visitors prefer the aesthetics of historic buildings, and these preferences are not dependent on gender, origin, or age. Presumably, the low values ascribed to the Lagoon resulted from the limited acquaintance of students with this site, which is located on the periphery of the area. Respondents provided few indications of landscape elements in this area and ascribed low rating positions to them. Only aesthetic values were appreciated. Furthermore, the implementation of the objective grading scale showed that the Lagoon possessed limited values. Among the study landscape objects located on the Campus, the most highly appreciated were educational and aesthetic values. The latter were generally ranked higher by the respondents than other intangible classes, regardless of the site type. This conclusion is consistent with the results of other papers [50,57], as scientific research generally shows that city visitors treat ULOs not in terms of their objective landscape properties, but in terms of their meaning and values. Therefore, in the assessment, they paid great attention to aesthetic values.

**Table 9.** Summary of results on differences among the social perception, expert assessment, and total values of analyzed sites.

|  | The Old Town | Campus | Lagoon | All Sites |
|---|---|---|---|---|
| Social perception | Good acquaintance of the site; high aesthetic and enjoyment values | Good acquaintance of the site; high aesthetic values | Limited acquaintance of the site | Aesthetic values were the most appreciated |
| Expert assessment | Outstanding heritage values; high entertainment and symbolic values | High educational values | Low heritage values | High entertainment and educational values |
| Intangible vs. tangible values | No differences | No differences | Intangible values are more significant | No differences |
| Total values (A$_{ULOs}$) | High | Medium | Low | - |
| ULO of the highest values |  Lublin Castle |  Main building of the John Paul II Catholic University of Lublin |  The bicycle path | Lublin Castle |
| ULO of the lowest values |  Ice cream shops near the castle |  Main building of Marie Curie-Sklodowska University |  The dam | The dam |

It must also be emphasized that the study results showed that the site with the highest level of anthropogenic transformation—the Old Town—possessed the highest values. By contrast, the most natural area—the Lagoon—featured the lowest values. Initially, the authors tried to explain this using the theory that a stronger sense of belonging to a given place leads to higher values attributed to it [58]. According to this statement, the Campus example, however, should possess the highest or at least similar intangible values to the Old Town, as respondents use the university infrastructure in their everyday life and, thus, their personal experience and connection to those objects should be strong. Another possible explanation relates to the multifunctional character of the Old Town. This site provides more services and more diverse sensations than other areas. As a result, its potential values are higher. This statement is consistent with the fact that people feel better being in the city center, because of access to commercial and cultural facilities whose aesthetics are usually high and fashionable. It is also worth mentioning the research of Keles et al. [59], who showed that the visual landscape quality assessment of historic city centers is dependent on the popularity of objects located in a given area.

*4.2. Advantages and Disadvantages of the Adopted Method*

The method we applied allowed us to (1) merge social and expert opinions on ULOs, (2) analyze diverse characteristics of urban space including tangible and intangible characteristics, (3) quantitatively present the result of values assessments enabling comparisons among different aspects, and (4) map both objective and subjective values of ULOs on the same scale (see Figure 5). Such an approach, based on both expert (universal grading scale) and social (PPGIS) opinions, could support urban landscape planning by providing an effective tool with which to obtain data featuring a high simplicity of structure (map) and that aids public understanding. To obtain a more comprehensive understanding, however, it would be necessary to collect the opinions of diverse social groups, not only those of students as in this pilot project, which was aimed at the introduction and testing of the method and not at the comprehensive analysis of values ascribed by different actor groups. We are aware of

the fact that the consideration of the opinions and perceptions of multiple users and beneficiaries is crucial as such an approach would likely mitigate conflicts in land use [60]. Furthermore, as stated, many studies [34,61] of different stakeholders' perceptions sometimes give more insight into the value concept than purely quantitative or monetary analysis.

We consider the biggest advantage of the method to be the presentation of the intangible characteristics of ULOs in a clear objective form, through the application of mathematical models reflecting the rank of a given element regarding its values. This is particularly important as, thanks to the adopted approach, tangible and intangible values can be expressed and presented graphically on the same scale. Our approach also allowed such elements to be ranked, as well as sites. The necessity of searching for methods that rate intangible values alongside tangible ones linked to a physical process was emphasized by other researchers [23,62]. The identification of sites generating the most or fewest benefits (using their ranking positions) is important from a managerial point of view [63,64], as places that were previously overlooked or treated marginally might turn out to possess many values when considered at the local level [65] and would then require different planning and management actions (e.g., protection or additional investment).

Another advantage of the proposed procedure derives from its high flexibility, meaning that it has potential to be applied in relation to different sites, regions, and countries. Such an approach is consistent with the European Landscape Convention [38], which emphasizes that all the research on landscape quality should be conducted in reference to areas of different types. Its application, however, may be affected by the differences among landscape types and operational units. For example, types of ULOs may vary among countries and cities. Some objects may not possess educational or spiritual values but be of great importance due to their impact on biodiversity or health services. It was partially shown by the conducted study that the framework used lacks the illusion of ecological values deriving from the existence of biologically active areas within the city structures. This underestimation was particularly seen in the case of the Lagoon possessing high ecological values which were not included in the assessment criteria. Moreover, PPGIS as used in the presented study may not be a traditionally used tool for collecting spatial information or may be practically infeasible. This is why it is an approach based on the merging of subjective and objective opinions and the aggregation method, which can be implemented in relation to different sites, regions, and countries. The ranking assessment criteria based on the four-point scale adopted in the assessment of tangible and intangible values of ULOs should be verified and adjusted to local environmental, social, and policy conditions.

Indeed, if ULO value assessments are to serve as a universal tool for spatial policy, their identification needs to be easy and fast. The use of a universal grading scale in reference to some of the value types significantly reduces labor intensity. The process of rank determination based on sociological studies, however, is time-consuming. Performing the social valorization via the Internet or sending questionnaires by mail can cause many difficulties [66]. It would be burdensome to find respondents who recently visited a given site; however, intangible values cannot be assessed without experiencing them directly [60,65]. As there is no method of sociological research that is simultaneously easy, fast, and effective, the application of PPGIS seems to be one of the most adequate solutions mitigating many of the problems dealing with traditional approaches [32,33]. In contrast to methods based solely on survey campaigns or geo-questionnaires, which are part of PPGIS [29,31], it objectively expresses both tangible and intangible values of ULOs using the same scale. It also gives the opportunity to reflect objective ratings on the map.

Regarding the future outlook, the conducted pilot study indicated further examination in reference to two aspects. First of all, future questionnaires on intangible features of ULOs should be directed to diverse groups of stakeholders. The city image can be different in different stakeholders' visions, such as the local community and visitors. This was proven by papers on Polish case studies [27,28], as well as in relation to other countries [26]. Taking into account the local condition, firstly, opinions of the following groups should be included in the model: employees of the city council, representatives of NGOs, ecologists, long-term residents, and tourists. Secondly, the evaluation framework should be

amplified to include a new layer on ecological values. These values may be reflected in an objective manner via the assessment of the percentage share of the biologically active areas within each site or via the assessment of the studies' ecological quality. The latter can be executed on the basis of the application of landscape metrics [67] or composite indices such as the indicator of ecological landscape quality [68] or index of landscape disharmony [69], proven to reflect landscape quality in relation to Polish environmental conditions. The subjective dimension of ecological values may be expressed twofold. First of all, by taking the opinion of ecologists in the first stage of the study, they probably would indicate ULOs of high ecological values, unknown to other groups of respondents. Secondly, ecologists' subjective assessment of the intangible values would give more scores to environmental values, which may be underestimated by other groups.

## 5. Conclusions

The applied method fulfilled all the requirement set out at the beginning of the study as it allowed us to (1) merge social and expert opinions on ULOs, (2) analyze diverse characteristics of urban space including tangible and intangible characteristics, (3) quantitatively present the results of value assessments enabling comparisons among different aspects, and (4) map both objective and subjective values of ULOs on the same scale. Additionally, the example test area enabled the authors to answer the three following research questions being the basis of the study development:

How do people perceive ULOs located in different sites? Respondents highly appreciated the Old Town and the Campus as they used ULOs located in these areas in their everyday life. Their personal experience and connection to those objects was stronger and more easily expressed. Respondents had only limited acquaintance with the Lagoon as they provided few indications of landscape elements in this area and ascribed low rating positions to them. This likely resulted from the peripheral localization of the Lagoon and the homogeneous characteristics of elements located within this site.

What kinds of tangible values are possessed by different ULOs and how can they be expressed? The results showed that the analyzed ULOs, depending on the site type, possessed outstanding heritage values, high entertainment, and high symbolic values (the Old Town), high educational values (Campus), and no outstanding values (the Lagoon). These values could be expressed on the basis of the objective grading scale, which was itself based on unequivocal and open data, available on official databases. As a result, the results of the assessment would be the same regardless of the evaluator.

How can intangible and tangible values be merged? The aggregation procedures based on the application of mathematical models allowed the authors to objectively express both tangible and intangible values of ULOs using the same scale and on a map. Such a form of presentation is of high relevance to decision-makers and is easily understood by the general public. In urban planning, spatial presentation is a basis for making decisions in reference to the real location of such areas.

To summarize, the proposed method has the potential to be implemented in the practice of urban landscape planning at the city level, providing the illusion of opinions from diverse actor groups and supplementing the method with criteria from ecological value assessments.

**Author Contributions:** Conceptualization, B.S.-Ś. and M.M.-Ś.; methodology, B.S.-Ś.; software, B.S.-Ś. and M.M.-Ś.; validation, B.S.-Ś.; formal analysis, B.S.-Ś.; investigation, B.S.-Ś., M.M.-Ś. and D.S.; resources, B.S.-Ś., M.M.-Ś., and A.K.; data curation, B.S.-Ś. and M.M.-Ś.; writing—original draft preparation, B.S.-Ś. and M.M.-Ś.; writing—review and editing, M.M.-Ś. and A.K.; visualization, M.M.-Ś.; supervision, B.S.-Ś.; project administration, M.M.-Ś.; funding acquisition, B.S.-Ś. and M.M.-Ś. All authors have read and agreed to the published version of the manuscript.

**Funding:** The research was funded by the Ministry of Science and Higher Education (Poland) for the dissemination of science (766/P-DUN/2019).

**Conflicts of Interest:** The authors declare no conflict of interest.

# Appendix A

**Table A1.** The mean grading points ascribed to analyzed ULOs and the results of the ranking and aggregation procedures.

| Area | Numerical Symbol | Urban Landscape Elements | Subjective Approach | | | | | Objective Approach | | | $A_{ULEs}$ |
|------|------|------|------|------|------|------|------|------|------|------|------|
| | | | Aesthetic | Spiritual | Enjoyment | Universal | Symbolic | Educational | Entertainment | Heritage | |
| The Old Town | 1 | Lublin Castle | 2.4 | 1.4 | 1.7 | 2.0 | 3 | 3 | 3 | 2 | 0.93 |
| | 2 | Stairs leading to the castle | 1.9 | 1.1 | 1.5 | 1.5 | 1 | 1 | 2 | 2 | 0.56 |
| | 3 | Castle Square | 1.8 | 0.9 | 1.4 | 1.4 | 2 | 2 | 3 | 0 | 0.73 |
| | 4 | Castle bridge | 2.2 | 1.2 | 1.7 | 1.7 | 2 | 1 | 0 | 0 | 0.48 |
| | 5 | Parkland next to the Old Town (Błonia) | 1.7 | 1.4 | 1.4 | 1.4 | 1 | 0 | 3 | 0 | 0.61 |
| | 6 | Ice cream shops near the castle | 1.1 | 0.6 | 0.9 | 0.8 | 1 | 0 | 0 | 1 | 0.10 |
| | 7 | Archcathedral | 2.4 | 2.3 | 1.5 | 1.8 | 3 | 2 | 3 | 2 | 0.92 |
| | 8 | Trinity Tower | 2.1 | 1.0 | 1.1 | 1.7 | 3 | 3 | 2 | 2 | 0.89 |
| | 9 | Crown Tribunal | 1.9 | 0.8 | 0.9 | 1.3 | 2 | 2 | 3 | 2 | 0.81 |
| | 10 | Cracow Gate | 2.3 | 1.0 | 1.8 | 1.9 | 3 | 3 | 1 | 2 | 0.82 |
| | 11 | Fara Square | 2.1 | 1.4 | 1.9 | 1.9 | 2 | 2 | 3 | 2 | 0.91 |
| | 12 | Pub "U Szewca" | 2.1 | 0.8 | 1.3 | 1.2 | 1 | 1 | 3 | 2 | 0.64 |
| | 13 | Restaurant "Czarcia łapa" | 1.9 | 0.8 | 1.0 | 1.1 | 0 | 1 | 1 | 1 | 0.27 |
| | 14 | Restaurant "Stół I wół" | 2.0 | 0.7 | 1.2 | 1.0 | 0 | 1 | 1 | 1 | 0.31 |
| | 15 | Dominican Church | 2.3 | 2.3 | 1.4 | 1.8 | 3 | 2 | 3 | 2 | 0.92 |
| | 16 | Pub "U Świętego Michała" | 2.1 | 0.7 | 1.3 | 1.0 | 0 | 1 | 1 | 2 | 0.40 |
| Campus | 17 | The building of "AGRO" | 2.0 | 1.2 | 1.0 | 1.3 | 1 | 3 | 3 | 0 | 0.66 |
| | 18 | Swimming pool | 2.0 | 0.8 | 1.4 | 1.6 | 1 | 2 | 3 | 0 | 0.64 |
| | 19 | The new building of veterinary department | 2.4 | 0.9 | 0.6 | 1.0 | 0 | 3 | 1 | 0 | 0.44 |
| | 20 | Centre of innovation and implementation | 2.6 | 1.2 | 1.2 | 1.6 | 0 | 3 | 1 | 0 | 0.55 |
| | 21 | Library | 2.1 | 2.0 | 1.4 | 1.8 | 0 | 3 | 2 | 0 | 0.57 |
| | 22 | Building of rector's office | 1.4 | 0.8 | 1.0 | 1.2 | 1 | 3 | 0 | 0 | 0.42 |
| | 23 | University Park | 1.7 | 1.4 | 1.6 | 1.6 | 0 | 1 | 1 | 0 | 0.44 |
| | 24 | Main building of Marie Curie-Skłodowska University | 1.0 | 0.7 | 0.4 | 0.8 | 1 | 3 | 1 | 0 | 0.20 |
| | 25 | Main building of the University | 2.5 | 1.7 | 1.0 | 1.5 | 2 | 3 | 3 | 0 | 0.77 |
| | 26 | Cinema "Bajka" | 0.9 | 0.6 | 0.7 | 1.0 | 0 | 1 | 3 | 0 | 0.27 |
| | 27 | The University sport center | 1.4 | 0.8 | 0.9 | 1.2 | 0 | 2 | 3 | 0 | 0.54 |
| | 28 | Students' dormitory of UMCS | 1.2 | 0.8 | 0.8 | 0.9 | 0 | 2 | 0 | 0 | 0.21 |
| Lagoon | 29 | Tennis courts | 1.8 | 0.9 | 1.2 | 1.3 | 0 | 0 | 3 | 0 | 0.47 |
| | 30 | Night club "Pause" | 0.7 | 0.4 | 0.9 | 0.8 | 0 | 0 | 3 | 0 | 0.14 |
| | 31 | Aqua park "Słoneczny Wrotków" | 1.6 | 1.2 | 1.0 | 1.0 | 0 | 0 | 3 | 0 | 0.42 |
| | 32 | The dam | 1.2 | 0.6 | 0.7 | 0.8 | 0 | 1 | 0 | 0 | 0.07 |
| | 33 | The marina | 1.7 | 1.5 | 1.2 | 1.7 | 1 | 1 | 1 | 0 | 0.45 |
| | 34 | Bicycle path | 2.1 | 1.4 | 1.5 | 1.4 | 1 | 1 | 1 | 0 | 0.45 |

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
