# Peer review of "In the Search of an Assessment Method for Urban Landscape Objects (ULOs): Tangible and Intangible Values, Public Participation Geographic Information Systems (PPGIS), and Ranking Approach"

_land, doi:10.3390/land9120502_

Round 1

Reviewer 1 Report

Dear all,

Thanks for the opportunity to review this manuscript.

In fact, the topic is pertinent and interesting. Also, the work is overall well conducted. Nevertheless, some revisions should be considered before the acceptance of this work, as:

  • In the title, PPGIS should be explained and not only the acronym
  • In the abstract, the most relevant results of this research should be added (also in numbers)
  • A section regarding the study limitations and prospective research lines should be added

Best regards,

Author Response

Response to Reviewer 1 Comments

Thank you very much for the review of our manuscript entitled: “In the search of Urban Landscape Objects’ (ULO) method of assessment: tangible and intangible values, PPGIS and ranking approach”. We sincerely appreciate all valuable comments and suggestions, which helped us to improve the quality of the article. Our responses to the comment are described below in a point-to-point manner. Appropriated changes has been introduced to the manuscript.

Point 1: In the title, PPGIS should be explained and not only the acronym

Response 1: We agree, PPGIS will be explained in the title

Point 2: In the abstract, the most relevant results of this research should be added (also in numbers)

Response 2: Due to the word limit (200 words) we were not able to add the most relevant results to the abstract. That is why, we decided to change the last sentence of the Abstract in a  following way:

The sentence: The aggregation procedures based on the application of mathematical models allowed authors to objectively express both tangible and intangible values of ULOs using the same scale and expressed them on a map. The proposed method has the potential to be implemented into the practice of urban landscape planning at the city level was removed from the Abstract.

The sentence was added to the Abstract: The general finding of the study showed that the Old Town features the highest ranked position in terms of all the values (mean AULOs=0.64), together with the higher share of the most appreciated ULOs, wherein the Lagoon the lowest (mean AULOs=0.35) also comparable with the campus (mean AULOs=0.45).

Point 3: A section regarding the study limitations and prospective research lines should be added

Response 3: Aspects of the study limitation has beeen included in a first version of the paper (Discussion section: 4.2. Advantages and disadvantages of the adopted method). They include:

(1) the illusion of one group of respondents (students); Lines  341-349;

(2) lacks of the illusion of ecological values; Lines 365-370;

(3) the difficlulty to find respondents who have recently visited a given site; Lines 377-379

A paragraph on prospective research was added at the end of section 4.2.; Lines 461-468

Reviewer 2 Report

This manuscript aims to develop and test an integrated method of assessment regarding the Urban Landscape Objects. The manuscript's hypothesis about the need to integrate the objective and subjective approaches in urban planning is very significant and supports the manuscript's novelty. However, to this reviewer, the manuscript needs further revision before it can be considered for publishing. Below, please find my comments on each section of the article.

General considerations:

(i). I think the state of the art can be better support the research hypothesis. It is essential to discuss what the current planning issues are in Lublin city? Why do the integration of the objective and subjective approaches supposed to be a smart solution for this city?

(ii) In terms of landscape assessment, the manuscript need to explain:

- If the current study has developed its value assessment framework based on relevant research? The used value framework is somehow weak and not proper. For instance, it is not clear how SPIRITUAL value has been inserted to the PPGIS method or whether the HERITAGE value is distinct from ENTERTAINMENT or SYMBOLIC. The authors are suggested to develop a robust theoretical framework to support the used variables.

- How the sampling and selection of participants support the diversity and credibility of the findings? The Burra Charter declares that "Places may have a range of values for different individuals or groups." However, in this study, the perception of value is dedicated to 80 students.

- How do you assess the value and integrate objective and subjective approaches in terms of institutional or community conflicts? For instance, inline 66, the authors discuss "the image of the city" as one of the main factors for determining the quality of life. However, the city image can be different in different stakeholders' visions, like the local community and visitors. Please see [Joining Historic Cities to the Global World: Feasibility or Fantasy?]

- Finally, the authors claim that their proposed method can be implemented into the practice of urban landscape planning at the city level by providing the illusion of opinions from diverse actor groups and supplementing the method with criteria from ecological value assessments. However, it is unclear how communication may be established between ecologists and urban planners, and what barriers can impact the assessment process?

Specific considerations:

 [37-38] do you believe globalization and rapid urbanization are drivers for quality of life and peoples' well-being across the European cities?

[41] … should be consistent with the concept of sustainable development. Please explain in which aspects? And how?

[44-45] so, what are differences and what are similarities?

[103-106] here is my main concern about the valorization framework. It needs to be strengthened based on supportive studies that suggest relevant determinants.

[151-152] Figure 2. I wonder how intangible values, as bottom-up approaches, can be generated without indicating the roles of NGOs and field works?

[227-236] Do you believe the result potentially can be changed if we select another group of participants (for instance, by age)?

[364-366] I advise avoiding such stereotypes in academic reasoning, as this framework's accuracy has not been endorsed in other landscapes yet. Instead, I would suggest expanding discussion about the barriers that can affect such an approach's efficiency in other places.

Author Response

Response to Reviewer 2 Comments

Thank you very much for the review of our manuscript entitled: “In the search of Urban Landscape Objects’ (ULO) method of assessment: tangible and intangible values, PPGIS and ranking approach”. We sincerely appreciate all valuable comments and suggestions, which helped us to improve the quality of the article. Our responses to the comment are described below in a point-to-point manner. Appropriated changes has been introduced to the manuscript – marked in red.

Point 1: I think the state of the art can be better support the research hypothesis. It is essential to discuss what the current planning issues are in Lublin city? Why do the integration of the objective and subjective approaches supposed to be a smart solution for this city?

Response 1: We thank the reviewer for pointing out the lack of justification of these issues in our article. We agree that it requires explanation and description. Integration of the objective and subjective approaches in landscape assessment, which is important element of spatial planning in the city, is a smart solution for city of Lublin, because of a few reasons. Firstly, the state of social dialogue and the level of public participation in all parts of Poland is low. Polish  law  allows  for participation  in  urban  planning  procedures,  but  not everyone wants  to exercise this right. This is due to the weakness of the NGO sector, inadequate local planning information sources and low activity of local communities (Jasiecki 2015, Feltynowski 2015). In all polish cities and  communes meetings and workshops where the opinions of residents are collected usually are organized in a specific place and time, so not everyone can participate in them. Our method allows to collect opinions and evaluations via the Internet. Therefore, all social groups can express their opinion. The presented method was used in Lublin, but it can also be applied to other cities in Poland and other parts of the world with similar problems of law social involvements.

Explanation was added in text: Introduction; Lines: 134-145

Point 2: In terms of landscape assessment, the manuscript need to explain:

- If the current study has developed its value assessment framework based on relevant research? The used value framework is somehow weak and not proper. For instance, it is not clear how SPIRITUAL value has been inserted to the PPGIS method or whether the HERITAGE value is distinct from ENTERTAINMENT or SYMBOLIC. The authors are suggested to develop a robust theoretical framework to support the used variables.

Response 2: Thank you for your comment. The theoretical background of the general structure of the valorisation framework  based on the merging of subjective and objective values was added in the Introduction based on the relevant references (also See Respond 9 and Introduction; Lines 111-127) 

As it was explained in the text (Method; Lines 184-219) the criteria for the values’ assessment were based on the systematic review on aspects related to urban objects’ values. The review comprised two-stages. Firstly, the English literature review was conducted based on Scopus database (2019) using keywords such as (cultural) ecosystem services, urban (city) assessment, urban (city) objects and urban (city) values to detect which values are of a tangible character and can be expressed objectively. Here, the Common International Classification of Ecosystem Services (CICES), version V5.1.  framework for the assessment of ecosystem services was the most. This Classification  was elected as has been widely used as the basis of mapping landscape values across different regions and countries (Haines-Young and  Potschin-Young https://doi.org/10.3897/oneeco.3.e27108)  and has strong legal and methodological basis as was developed out of the work on environmental accounting undertaken by the European Environment Agency (EEA). This classification divided cultural values into two general groups: (1) Direct, in-situ and outdoor interactions with living systems that depend on presence in the environmental setting, which are considering as intangible values in our study.; and (2) Indirect, remote, often indoor interactions with living systems that do not require presence in the environmental setting which are considering as tangible values in our study. Among them are mentioned;

  • Characteristics (biotic) that enable active physical interactions and characteristics that enable activities promoting health, recuperation or enjoyment, elements used for entertainment (biotic) -  assessed as  enjoyment intangible values and entertainment intangible values
  • Characteristics (abiotic) that enable intellectual interactions and characteristics that enable education and training - assessed as educational intangible values
  • Characteristics or features (abiotic) that have either an existence, option or bequest value and characteristics (biotic) that are resonant in terms of culture or heritage - assessed as universal intangible values and heritage intangible values
  • Spiritual interactions (abiotic) and elements (biotic) that have sacred or religious meaning – assessed as spiritual intangible values
  • Characteristics (abiotic) that enable symbolic interactions and elements (biotic) that have symbolic meaning – assessed symbolic intangible values
  • Characteristics (abiotic) that enable active or passive experiential interactions and characteristics (biotic) that enable aesthetic experiences – assessed as aesthetic intangible values

The  detailed criteria of values assessment (questionnaire structure and objective grading scale) were partially based on the results of the review of literature  (SkÅ™ivanová, Z.; Kalivoda, O. Perception and assessment of landscape aesthetic values in the Czech Republic – a literature review. Journal of Landscape Studies. 2010.  3, pp. 211 – 220.; Oladeji, S.O.; Agbelusi, E.A.; Ajiboye, A.S. Assessment of Aesthetic Valeus of Old Oyo National Park. American Journal of Tourism Management. 2012, 1(3), pp. 69-77 DOI: 10.5923/j.tourism.20120103.02 Subiza-Péreza, M.; Vozmediano, L.; San Juana, C. Green and blue settings as providers of mental health ecosystem services: Comparing urban beaches and parks and building a predictive model of psychological restoration. Landscape and Urban Planning 2020. 204.  https://doi.org/10.1016/j.landurbplan.2020.103926).  The final structure of the framework derives from the local context: the availability/ lack of data for objective criteria and the characteristic of ULOs detected by respondents.

Point 3: In terms of landscape assessment, the manuscript need to explain:

How the sampling and selection of participants support the diversity and credibility of the findings? The Burra Charter declares that "Places may have a range of values for different individuals or groups." However, in this study, the perception of value is dedicated to 80 students.

Response 3:  We agree with the reviewer that places may have a range of values for different individuals or groups and we are aware of this fact at the time of paper preparation. The presented study, as it is explained in the goal of the study  (the main aim of the paper is to develop and test the method of ULOs assessment) and discussions section (Lines 388-399), constituting a pilot study aiming at the test of the methodology. It’s aim is not to analyse diversities among stakeholders views and opinions or present a whole spectrum of subjective values attributed to ULOs.  As we emphasised in the new paragraph added to the discussion,  to obtain such a whole picture diverse groups of stakeholders should be included in future research. They primary are: employees of the city council, representatives of NGOs, long-term residents and tourists.

Point 4: In terms of landscape assessment, the manuscript need to explain:

How do you assess the value and integrate objective and subjective approaches in terms of institutional or community conflicts? For instance, inline 66, the authors discuss "the image of the city" as one of the main factors for determining the quality of life. However, the city image can be different in different stakeholders' visions, like the local community and visitors. Please see [Joining Historic Cities to the Global World: Feasibility or Fantasy?]

Response 4:  

We agree with the reviewer that city image can be different in different stakeholders' visions, like the local community and visitors. These was proven by papers on Polish case studies (Osóch B, CzapliÅ„ska A. City image based on mental maps — the case study of Szczecin (Poland). Miscellanea Geographica 2019, Volume 23: Issue 2, p DOI: https://doi.org/10.2478/mgrsd-2019-0016.; Huang J., Obracht-ProndzyÅ„ksa H et al. 2021. The image of the City on social media: A comparative study using “Big Data” and “Small Data” methods in the Tri-City Region in Poland. Landscape and Urban Planning Vol 26. Doi: https://doi.org/10.1016/j.landurbplan.2020.103977) as well as in relation to other countries (Dastgerdi A. S.; De Luca, G. Joining Historic Cities to the Global World: Feasibility or Fantasy? Sustainability 2019. 11(9), 2662; https://doi.org/10.3390/su11092662). The explanation of this fact will  be added to the paper. In relation to presented study, to avoid  institutional and community conflicts,  objective criteria of assessment were based on: (1) widely accepted issues such as the need of the protection of culturally important sites and objects (e.g. Heritage values); (2)  the historically confirmed importance of some objects (e.g. Symbolic values). To obtain the consensus among all the stakeholders groups, however, it is  crucial to include in the subjective assessment on ULOs opinion of managers, local communities and visitors – see response to Point 3.

Explanation was added in text: Discusssion; Lines 452-456

Point 5: The authors claim that their proposed method can be implemented into the practice of urban landscape planning at the city level by providing the illusion of opinions from diverse actor groups and supplementing the method with criteria from ecological value assessments. However, it is unclear how communication may be established between ecologists and urban planners, and what barriers can impact the assessment process?

Response 5:  We agree with the reviewer that the aspect of  communication between ecologists and urban planners should be discussed.  In Polish conditions, city planning is typically focused on coherently organizing city systems dealing with the planning of LU types and functions. Parks, green and open spaces are usually a part of urban plans, however, ecology and biological process are on the sidelines. The main problem of Polish cities is urban sprawl, mainly resulted from the lack of local spatial plans. Jasiecki, K. Social participation problems in Poland and the way they influence public policy. Studia z Polityki Publicznej (Public Policy Studies). 2015. 3(7) pp. 101-119, Feltynowski, M. Public Participation in Spatial Planning in Poland as an Element of Evidence Based Urban Planning – Case Study of Lodz. Journal of European Economy 2015. 14(3), pp. 280-289. As a result, green areas located next to cities are gradually covered by buildings and paved roads.  Here is the main communication barrier between urban planners and ecologists. The actions of the first group lead in most cases to a decrease in the share of green areas: economic and functional factors (efficient transport corridor) and the pressure of investors win against environmental needs. While the second group most often presents a conservative approach to the development of green areas. That is why, it is crucial to include ecologists as one of the consultants’ groups  in urban planning and design. Presented method gives such as opportunity by including ecologists as one of the groups of respondents. They contribution will be have a twofold impact. First of all, in the first stage of the study they probably would indicate ULOs of high ecological values, unknown to other respondents groups. Secondly,  their subjective assessment of the intangible values would give more scores to ecological values, which may be underestimated by other groups. That is why ecologists, next to employees of the city council, representatives of NGOs, long-term residents and tourists, will be included in future analysis based on the application of the framework presented in the paper.

Explanation was added in text: Discusssion; Lines 452-468

Point 6: Line 37-38 do you believe globalization and rapid urbanization are drivers for quality of life and peoples' well-being across the European cities?

Response 6: The statement in the introduction does not fully reflect the sense of our reasoning. Thank you for drawing your attention to it. We meant that despite such a large population growth in cities, it is expected that they will provide a decent quality of life and contribute to improving people's well-being. Will be changed in the text: Lines: 35-39

Point 7: Line 41 … should be consistent with the concept of sustainable development. Please explain in which aspects? And how?

Response 7: The valuation of the urban landscape should be consistent with the concept of sustainable development, especially in terms of preserving cultural landscapes as well as directing and harmonizing its changes resulting from social, economic and environmental processes. This valuation should simultaneously draw upon social impact assessments, by involving the community in decision-making processes. Completed in the text: Lines: 41-42

Point 8: Line 44-45 so, what are differences and what are similarities?

Response 8: We believe that the reviewer meant the differences in the methods of assessing urban spaces. Therefore these methods differ mainly from the point of view of the type of assessed urban space. However, when it comes to the similarities, considering them would be a topic for a separate article. What is why we only mentioned the main areas of similarities in the text, Lines: 53-55

Point 9: Line 103-106 here is my main concern about the valorization framework. It needs to be strengthened based on supportive studies that suggest relevant determinants.

Response 9: The methodological framework together with the goal of research were strengthen by the references on the need to merge subjective and objective point of view in the urban landscape quality assessment. They include:  SowiÅ„ska-Åšwierkosz, Chmielewski, 2016; ELC 2000;  Plieninger et al. 2013; Mitsche et al. 2013; Siountri et al. 2019; Kalaycı Önaç and BiriÅŸçi, 2019; Wang et al. 2017;  Gundor and Polat 2018; Hems 2006; Wills-Herrera et al 2008.

Explanation was added in text: Introduction; Lines 111-127

Point 10: Line 151-152 Figure 2. I wonder how intangible values, as bottom-up approaches, can be generated without indicating the roles of NGOs and field works?

Response 10:  As it was explained in the Response 3 and new version of discussion section, presented research is a pilot study. The role of NGOs will be included by examining them as one of the crucial respondents groups in further research. Regarding the role of field works, in broad meaning they were conducted by both expert and respondents. Authors (Experts who ascribed grading points to each landscape element) conducted other field research in relation to Lublin case study and many times have been visited analysed sites. This ‘field knowledge’ helped to formulate questions and select criteria of assessment as well as enable us to proper interpret the results. Besides, respondents (students) had to be personally at analysed sites to be able to took part in the study. They ‘field knowledge’ was necessary to express subjective values to analysed ULOs.

Point 11: Line 227-236 Do you believe the result potentially can be changed if we select another group of participants (for instance, by age)?

Response 11: Yes, we think that there is a possibility of such a change which was partially proven by the study results. The fact that students pointed only four location on the Lagoon site resulted from their  limited acquaintance with this site, which is located on the periphery of the area (this issue was mentioned in the discussion section) . Inversely, they pointed quite a lot of object within the Campus (72) being the place of their everyday life. Ones again we want to emphases that the paper has a  form of pilot study aiming at the test of method and mathematical model, not at the full assessment opinions of different respondents groups’.

Point 12: Line 364-366 I advise avoiding such stereotypes in academic reasoning, as this framework's accuracy has not been endorsed in other landscapes yet. Instead, I would suggest expanding discussion about the barriers that can affect such an approach's efficiency in other places.

Response 12:

Thank you for your comment. We corrected this sentence in the text to: Another advantage of the proposed procedure derives from its high flexibility, meaning that has a potential to be applied in relation to different sites, regions and countries.

Besides a section of the barriers that can affect such an approach's efficiency was added: Discussion;  Lines 461-468

Round 2

Reviewer 2 Report

The revised manuscript has responded to all the comments and concerns proposed by this reviewer very satisfactorily. Thanks to the authors, they made an impressive revision.